# Isolation of a T7-Like Lytic *Pasteurella* Bacteriophage vB_PmuP_PHB01 and Its Potential Use in Therapy against *Pasteurella multocida* Infections

**DOI:** 10.3390/v11010086

**Published:** 2019-01-21

**Authors:** Yibao Chen, Guanghao Guo, Erchao Sun, Jiaoyang Song, Lan Yang, Lili Zhu, Wan Liang, Lin Hua, Zhong Peng, Xibiao Tang, Huanchun Chen, Bin Wu

**Affiliations:** 1State Key laboratory of Agricultural Microbiology, College of Animal Science and Veterinary Medicine, Huazhong Agricultural University, Wuhan 430070, China; yibaochen@webmail.hzau.edu.cn (Y.C.); e8348389@163.com (G.G.); erchaosun@163.com (E.S.); jiaoyangsong@163.com (J.S.); m15271936513@163.com (L.Y.); 13397173557@163.com (L.Z.); liangwan521521@163.com (W.L.); hualin0@webmail.hzau.edu.cn (L.H.); chenhch@mail.hzau.edu.cn (H.C.); 2The Cooperative Innovation Center for Sustainable Pig Production, Huazhong Agricultural University, Wuhan 430070, China; tangren77@163.com; 3Key Laboratory of Prevention and Control Agents for Animal Bacteriosis (Ministry of Agriculture), Animal Husbandry and Veterinary Institute, Hubei Academy of Agricultural Science, Wuhan 430064, China

**Keywords:** bacteriophage, lytic, *P. multocida* type D, isolation, therapeutic application

## Abstract

A lytic bacteriophage PHB01 specific for *Pasteurella multocida* type D was isolated from the sewage water collected from a pig farm. This phage had the typical morphology of the family *Podoviridae*, order *Caudovirales*, presenting an isometric polyhedral head and a short noncontractile tail. PHB01 was able to infect most of the non-toxigenic *P. multocida* type D strains tested, but not toxigenic type D strains and those belonging to other capsular types. Phage PHB01, the first lytic phage specific for *P. multocida* type D sequenced thus far, presents a 37,287-bp double-stranded DNA genome with a 223-bp terminal redundancy. The PHB01 genome showed the highest homology with that of PHB02, a lytic phage specific for *P. multocida* type A. Phylogenetic analysis showed that PHB01 and PHB02 were composed of a genus that was close to the T7-virus genus. In vivo tests using mouse models showed that the administration of PHB01 was safe to the mice and had a good effect on treating the mice infected with different *P. multocida* type D strains including virulent strain HN05. These findings suggest that PHB01 has a potential use in therapy against infections caused by *P. multocida* type D.

## 1. Introduction

*Pasteurella multocida* isolates are generally classified into five capsular types (A, B, D, E, and F) [1], and are commonly associated with respiratory diseases and hemorrhagic septicemia in a wide range of domestic and wild animals [2]. While it rarely occurs, *P. multocida* infections in humans have been continuously reported [3,4,5,6,7], and most of these cases are likely to be transmitted from pets such as dogs and cats [4,8]. It has been reported that the infection of *P. multocida* isolates displays host predilection; different capsular types are associated with specific types of diseases [2,8,9,10].

Known as the natural predators of bacteria, bacteriophages (phages) are probably the most abundant biological entity on the Earth [11]. Regarding their ability to kill pathogens with high specificity, phages are proposed as promising therapeutic tools, and the use of phages/phage-derivatives to combat bacterial infections has been widely studied for bacteria such as *Escherichia coli*, *Pseudomonas aeruginosa*, *Klebsiella pneumoniae*, and *Clostridium difficile* [12,13,14,15]. However, current knowledge on *P. multocida* phages, especially the lytic phages, is limited.

*P. multocida* phages were first reported in 1956 [16], however, the lytic *P. multocida* phages had not been characterized until 2018 [17,18]. To date, there have been only five whole genome sequences of *P. multocida* phages available in the GenBank database including the three temperate transducing *Pasteurella* phages F108 (Accession: NC_008193) [19], AFS-2018a (Accession: MH238466), and Pm86 (Accession: MH238467) [20], and two lytic phages PMP-GADVASU-IND (Accession: KY203335) [18], and PHB02 (Accession: MF034659) [17]. While lytic phages specific for *P. multocida* capsular types A and B have been characterized recently [17,18], there is still a lack of information on lytic phages for other *P. multocida* capsular types. In this study, a novel lytic T7-like phage designated vB_PmuP_PHB01 (hereafter referred to as phage PHB01) specific for *P. multocida* capsular type D strains was isolated and characterized. To the best of our knowledge, this was the first report of a T7-like phage specific for *P. multocida* capsular type D strains.

## 2. Materials and Methods

### 2.1. Phage Isolation

PHB01 was isolated from the sewage water collected from a pig farm in Hubei Province in January 2016 using the conventional double-layer agar method described previously [17,21]. Briefly, 20 mL of sewage water was sterilized by filtration through a 0.22 μm pore size membrane. After that, 5 mL of the filtrate was mixed with 10 mL of indicator bacteria (*P. multocida* capsular type D strain HND065, a nontoxigenic isolate) at mid-log phase and incubated at 37 °C for 4 h. The mixture was then centrifuged at 12,000× *g* for 10 min, and filtered using a 0.22 μm pore size membrane. A total of 100 μL of the supernatant was then mixed with 300 μL of the indicator bacterium, and was poured into 6 mL of molten soft Tryptic Soy Broth (TSB medium with 0.7% *w*/*v* agar) containing 10% *v*/*v* of newborn calf serum (NBS). Finally, the mixture was poured onto a prepared Tryptic Soy Agar (TSA medium with 1.5% *w*/*v* agar; SA top agar) containing 10% *v*/*v* of NBS and incubated overnight at 37 °C to numerate the plaques.

After the plaques were numerated, a single plaque was picked and re-suspended in 6 mL of a modified SM buffer [5.8 g of NaCl, 2.0 g of MgSO_4_·7H_2_O, 50 mL of Tris-HCl (pH 7.4), 5.0 mL of 2% gelatin] [22] for 3 h. The phage-containing SM buffer was then centrifuged at 12,000× *g* for 30 s and the supernatant was filtered through a 0.22 μm pore size membrane. After that, the phage preparations were given serial 10-fold dilutions with sterile SM buffer. Phage isolation by the double-layer agar method was repeated four more times, and the phage suspensions were stored at 4 °C. The phages were purified by CsCl gradient ultra-centrifugation, as described previously [17].

### 2.2. Electron Microscopy

The morphology of the phages was determined using a 100-kV transmission electron microscope (HITACHI H-7650, Tokyo, Japan) with the same protocol previously described [17]. The phage filtrate was stained negatively with 2% uranyl acetate after the addition of a drop of a phage suspension onto a grid surface, and the excess stain was removed immediately.

### 2.3. Thermolability and pH Sensitivity

The thermolability and pH sensitivity of PHB01 was tested as previously described [23] with minor modifications. The purified phage particles were given incubation at different temperatures (4 °C, 20 °C, 40 °C, 50 °C, 60 °C, and 70 °C) for 1 h to test the thermolability. Incubations at 37 °C for 1 h under different pH levels (2, 3, 4, 5, 6, 7, 8, 9, 10, 11, and 12) were set to test the pH sensitivity of the phage. Each assay was performed in triplicate. Samples were titered by the double-layer agar plate method [17].

### 2.4. One-Step Growth Curve

To test the one-step growth curve, PHB01 was mixed with its indicator bacteria (mid-log phase) at multiplicity of infection (MOI) of 0.001 and incubated at 37 °C for 5 min. After incubation, unabsorbed free phages were removed by centrifugation at 12,000× *g* for 30 s. The pellets were washed using pre-warmed TSB (37 °C) first, and then, the suspension was transferred to 20 mL of TSB followed by incubation at 37 °C [24]. From this moment (t = 0 min), a 0.5-mL sample was collected every 10 min for 90 min. A double-layer agar method [17] was used to determine the titration of the phage particles. The experiment was repeated three times. The latent period was followed by a single burst of phages, where the burst size was the average number of phages released per infected host cell and calculated as the ratio between the number of phages before and after the burst [25].

### 2.5. Host Range

Spot tests were performed to determine the host specificity of phage PHB01, in accordance with previous study [26]. A total of 48 *P. multocida* clinical isolates collected from different locations in China including 37 capsular type D strains, 10 capsular type A strains, and 1 capsular type F strain as well as other bacterial species including *Escherichia coli*, *Salmonella enterica* serovar *typhimurium*, *Salmonella enterica* serovar *choleraesuis*, and *Bordetella bronchiseptica* were used. Summarily, bacterial strains were grown to mid-log phase at 37 °C, and 300 μL of each bacterial culture was added into 3 mL of molten SA top agar. After each overlay solidified, 4 μL of the phage lysate (1 × 10^9^ pfu/mL) was spotted onto the bacterial overlays, dried, and then incubated at 37 °C for 8 h. The same volume of sterile phage buffer was also spotted onto the bacterial overlays and incubated under the same conditions as the controls. Lytic specificity was defined based on the formation of bacteriophage plaques. The spot tests were repeated three times to confirm the results. The efficiency of plating (EOP) values were determined by calculating the ratio of pfus of each phage-susceptible strain to the pfus of the indicator strain (*P. multocida* HND065). This experiment was repeated three times [17].

### 2.6. DNA Extraction and Analysis of Genome Sequence

The genomic DNA of PHB01 was extracted using the phenol-chloroform method [25]. Briefly, the purified phages were treated by proteinase K (100 mg/mL), SDS (10%, *w*/*v*), and EDTA (0.5 mM, pH 8.0) at 56 °C in water for 2 h. After that, the sample was washed three times by using an equal volume of mixture composed of phenol, chloroform, and isoamyl alcohol (25:24:1), followed by centrifugation at 4 °C, 12,000× *g* for 10 min, to remove the debris. In the next step, the supernatant was mixed with isoamyl alcohol kept at −20 °C overnight. The air-dried precipitate was washed three times with cold 75% ethanol, and the phages’ genomic DNA was finally dissolved in TE buffer (10 mM Tris-HCl, 1 mM EDTA [pH 8.0]).

Whole genome sequencing was performed at BGI (Shenzhen, China) on an Illumina HiSeq 2500 sequencer with 2 × 100 bp read length [17]. Libraries with an insert size of 270 bp were constructed using the NEBNext®Ultra™ II DNA Library Prep Kit (NEB, Ipswich, MA, US). Raw reads with low quality were filtered and eliminated by SOAPnuke (version 1.5.0) software (https://github.com/BGI-flexlab/SOAPnuke) [27] according to the following criteria: reads with a certain proportion of low quality (20) bases (40% as the default, parameter setting at 20 bp), and/or with a certain proportion of Ns (10% as the default, parameter setting at 1 bp) were removed. Adapter contamination (15 bp overlap between the adapter and reads as the default, parameter setting at 15 bp) and duplication contamination were also removed. The high-quality reads were then de novo assembled into the genome by means of SOAPdenovo2.04 [28,29]. The terminal sequences of the virus genome were determined by a modified statistical method, as described previously [30]. Glimmer 3.0 [31], Tandem Repeat Finder 4.09 [32], and tRNAscan-SE 2.0 [33] were used to predict protein-encoding putative open reading frames (ORFs), Tandem repeats, and transfer RNAs (tRNAs) encoded by the PHB01 genome, respectively, with the default parameters. The PHB01 whole genome sequence (WGS) and its annotations were finally deposited into GenBank under accession number MF166859. When required, a sequence comparison was performed using Easyfig v.2.0 [34]. Phylogenetic trees were constructed using MEGAX [35] with 1000 Bootstrap replications.

### 2.7. Animals and Ethic Statement

BALB/c mice (5-week-old) used in this study were purchased from the Huazhong Agricultural University Laboratory Animal Center (Wuhan, China). All animal tests performed in this study followed the Guide for the Care and Use of Laboratory Animals of Hubei Province (approved by the Standing Committee of the People’s Congress of Hubei Province) and were approved by the Ethical Committee for Animal Experiments at Huazhong Agricultural University, Wuhan, China. The approved number is HZAUMO-2018-023.

### 2.8. Safety Test

To study the toxicity of PHB01, six 5-week-old female BALB/c mice were randomly divided into two groups with three mice in each group. Mice in each group received an intraperitoneal injection of 100 μL PHB01 (10^9^ pfu/mL), and 100 μL PBS buffer, respectively. All mice were housed under the same conditions and were observed for seven days.

The health status of each group of mice was recorded by giving different scores (0: dead; 1: near death; 2: exudative accumulation around partially closed eyes; 3: lethargy and hunched back; 4: decreased physical activity and ruffled fur; 5: normal health, condition unremarkable), as previously described [36]. The total score for each group was recorded at least three times a day. At seven days post-injection, all mice were euthanized, and the livers, spleens, kidneys, and lungs were collected for histological examination. The data describing the health status of each group of mice are presented as the mean ± SD.

### 2.9. Mouse Infection and Treatment

A non-toxigenic *P. multocida* capsular type D isolate HND065 and a non-toxigenic *P. multocida* capsular type D virulent isolate HN05 that we had sequenced previously [9] were used to generate the mouse-infection model in this study. Before study, the minimum lethal dose (MLD) of each isolate on mice was determined as described previously [37]. After that, thirty-six 5-week-old female BALB/c mice were divided into six groups (I–VI) and each group contained six mice. Mice in Groups I~II and groups IV~V were challenged intraperitoneally with HND065 and HN05 at 2 × MLD (2 × 10^7^ CFU for HND065 and 3.2 × 10^4^ CFU for HN05) while mice in Groups III and VI received a challenge of PHB01 (10^8^ PFU) and PBS through the same routine, respectively. In the next step, each of the mice in *P. multocida*-challenged groups were given a treatment of PHB01 at 10^8^ PFU (Groups II and V) or PBS (Groups I and IV) at 6 h following bacterial inoculation, and then twice a day for five days (Figure 1). All mice were housed under the same conditions. Health statuses were monitored and recorded at least three times a day for 21 days. Scores were given as above-mentioned and the data describing the health status of the mice were expressed as the mean ± SD. Survival was analyzed using the Kaplan–Meier analysis with a log-rank test (statistically significant at *p* < 0.05).

## 3. Results

### 3.1. Morphological Characteristics

Using a *P. multocida* capsular type D strain HND065 as the indicator, a *Pasteurella* bacteriophage designated vB_PmuP_PHB01 (PHB01 for short) was isolated from the pig farm sewage water through the double-layer agar method [21]. After incubation and purification, PHB01 formed round transparent plaques with a clear boundary in the double agar. The plaques were approximately 0.5–1.5 mm in diameter with a surrounding halo (2–3 mm in diameter; Figure 2A). Electron microscopy showed that PHB01 had an isometric polyhedral head approximately 55 nm in diameter and a short tail ~13 nm in length (Figure 2B). Based on these morphological characteristics and according to the latest International Committee on Taxonomy of Viruses (ICTV) classification, PHB01 was determined as a member of the subfamily *Autographivirinae*, family *Podoviridae*, and the order *Caudovirales*.

### 3.2. Life Cycle Parameters

Thermolability tests showed that PHB01 was stable from 4 to 40 °C, but showed a titer reduction from 50 to 60 °C; moreover, the titer dramatically decreased (approximately 4.8 log units) from 60 to 70 °C (Figure 3A). For pH sensitivity, PHB01 was stable from pH 5.0 to 9.0, but the titer dropped approximately 3 and 2 log units at pH 3.0 and 11.0, respectively (Figure 3B). The one-step growth curve determination test showed that the entire life cycle of PHB01 consisted of an approximately 70-min infection process and an approximately 10-min eclipse period; the average burst size was 190 phage particles per infected cell after 70 min at 37 °C (Figure 3C).

### 3.3. Host Range of PHB01

Host range tests showed that PHB01 was able to lyse most of the *P. multocida* capsular type D isolates (22 out of 37) tested; all of these sensitive isolates were non-toxigenic. A small number of non-toxigenic capsular type D isolates (4 out of 37) displayed resistance to PHB01 (Table 1). It is quite strange that PHB01 had no effect on the capsular type D isolates which produce toxin (Table 1). In addition, PHB01 also did not show activity on lysing *P. multocida* capsular type A and F isolates as well as the bacteria in other genus (*E. coli*, *Salmonella* spp., and *B. bronchiseptica*, Table 1).

### 3.4. Genomic Characteristics of PHB01

PHB01 possessed a linear double-stranded DNA genome with a size of 37,287 bp in length, with a 223-bp terminal redundancy. The average G + C content of the genome sequence was approximately 40.7%. Prediction using Glimmer 3.0 identified 43 putative ORFs; of which 24 ORFs were assigned putative functions and the remaining 19 ORFs were annotated as hypothetical proteins (8 ORFs) and proteins with unknown functions (11 ORFs) (Appendix A). The proteins with known functions participated in DNA packing and morphogenesis, lysis, replication, and regulation (Appendix A). There were no transfer RNAs (tRNA) or other small RNA genes detected in the PHB01 genome sequence using tRNAscan-SE 2.0.

Phylogenetic analyses using the sequences of either the major capsid protein (Figure 4A), the DNA polymerases (Figure 4B), and/or the RNA polymerases (Figure 4C) showed that PHB01 belonged to subfamily *Autographivirinae*, and was closest to phage PHB02.

Sequence comparison showed that the average nucleotide identity between the genomes of PHB01 and PHB02 was 96.16% (calculated by ANI, http://enve-omics.ce.gatech.edu/ani/). PHB01 possessed a similar genetic structure as PHB02; most proteins encoded by the PHB02 genome were seen in PHB01 (Figure 5). With the exception of several hypothetical or function-unknown proteins, only a tail fiber protein (gp17) displayed a level of difference between the two phages (Figure 4). Compared to the tail fiber protein encoded by PHB02, the protein encoded by PHB01 had an insertion of 18 amino acids at the N-terminal and multiple amino acids deletion and change at its C-terminal (Appendix A).

### 3.5. Acute Toxicity

After an injection of PHB01 at a dose of 1.0 × 10^8^ PFU, the mice did not show any abnormal behavior and/or appearance when compared to the control mice (Figure 6A). The amounts of PHB01 in livers (pfu/g), lungs (pfu/g), spleens (pfu/g), kidneys (pfu/g), and blood (pfu/mL) of the injected mice decreased as time passed; no phages were detected in the lung of the injected mice at 72 h post injection (Figure 6B). In addition, none of the mice injected with the phage showed any changes in the eosinophils or basophils or other pathological changes among the main organs when compared with the control mice (Figure 7).

### 3.6. Therapeutic Effect of the Phage

To test the therapeutic effect of PHB01, we challenged mice with both the host strain of PHB01 and a non-toxigenic *P. multocida* capsular type D virulent isolate HN05 and then treated the infected mice with either the phage or PBS (Figure 1). Challenged mice treated with PBS (Groups I and IV) showed severe clinical signs (Figure 8A). Over 80% of the mice in these two groups died within five days post challenge (Figure 8B). In contrast, those challenged-mice treated with PHB01 (Groups II and V) showed much milder signs and a significantly increased survival rate (*p* ≤ 0.05) (Figure 8A,B). There were no observed clinical signs and no deaths recorded for the PHB01 (Group III) and/or PBS (Groups VI) control groups (Figure 8A,B).

## 4. Discussion

It has been more than 135 years since *P. multocida* was first shown to be the causative agent of fowl cholera by Louis Pasteur in 1881, and many studies on *P. multocida* have been published. However, the history on the study of *P. multocida* phages is relatively short, and only a few reports regarding *P. multocida* phages are available. To the best of our knowledge, the first paper involved in the *P. multocida* phage was published in 1956 [16,37], but it was not until 2006, when the first temperate transducing phage for *P. multocida*, designated F108, was sequenced and its whole genome sequence was published [19]. Since then, there have been no reports on *P. multocida* phages for 12 years. In 2018, the first complete genome sequences of lytic bacteriophages for *P. multocida* capsular types A and B were reported [17,18]. However, there is still a lack of reporting on lytic bacteriophages for *P. multocida* strains of other capsular types (D, E, and F). In the present study, the lytic bacteriophage specific for *P. multocida* capsular type D was isolated, characterized, and sequenced for the first time, which helps us understand more about *P. multocida* phages.

It has been reported that more than 95% of all phages described in the literature belong to the order *Caudovirales* (tailed ds-DNA phages), which comprises three families: the families *Myoviridae* (viruses with contractile tails), *Siphoviridae* (viruses with long, noncontractile tails), and *Podoviridae* (viruses with short noncontractile tails) [18,38]. Viruses in different families have icosahedral or oblate heads, but differ in the length and contractile abilities of their tails [38]. Electron microscopy showed that the appearance of PHB01 was composed of an isometric polyhedral head approximately 55 nm in diameter and a short tail ~13 nm in length (Figure 2B). These morphological characteristics are quite similar to phage PHB02, a lytic *P. multocida* phage belonging to the family *Podoviridae* in the order *Caudovirales* [17], but differ from the reported *Pasteurella* phages in family *Siphoviridae*, which have long noncontractile tails (≥100 nm) [18,39,40] and/or in the family *Siphoviridae* with long, contractile tails (≥100 nm) [19]. These findings suggest that PHB01 is a member of the family *Podoviridae* in the order *Caudovirales.*

PHB01 and PHB02 are also genetically and phylogenetically related. Phylogenetic analysis using the sequences of either the major capsid protein, the DNA polymerases, and/or the RNA polymerases showed that PHB01 was closest to PHB02 (Figure 4A–C). Comparative genomics analysis showed that PHB01 shared a similar genetic structure to PHB02; most of the function-known proteins encoded by PHB01 and PHB02 were the same, with the exception of the tail fiber protein (gp17) (Figure 5). It has been reported that *Caudovirales* phages use the tail fiber protein to recognize and attach to the bacterial surface at the early stage of infection by specifically digesting the polysaccharides, the primary receptors for phages [41,42,43,44,45]. It is also known that PHB01 and PHB02 are specific for *P. multocida* capsular type D and A strains, respectively [18], and *P. multocida* strains of capsular type D and A produce different polysaccharides [46]. Therefore, the different tail fiber proteins encoded by PHB01 and PHB02 might be associated with the host specificity of the two phages. Only when more phage genomes from different *P. multocida* capsular types become available can we finally confirm this suggestion.

Host range tests showed that PHB01 displayed good specificity for killing *P. multocida* capsular type D isolates (Table 1). However, some type D isolates, particularly the toxigenic type D isolates, were highly resistant to PHB01 (Table 1). It has been documented that bacteria resist phages by preventing phage adsorption, preventing phage DNA entry, cutting phage nucleic acids, or by a wide range of heterologous proteins that provide resistance through the abortion of phage infection (known as the abortive infection systems) [47]. In the next step, we will undertake further study to find out about such mechanisms in those type D isolates. In vivo tests using mouse models showed that the administration of PHB01 was safe for the mice and had a good effect on treating the mice infected with different *P. multocida* strains including virulent strain HN05 (Figure 8A,B). These findings suggest that PHB01 has a potential use in therapy against *P. multocida* infections.

## 5. Conclusions

A novel lytic bacteriophage PHB01 specific for *P. multocida* type D was characterized in the present study. Both morphological and genetical analysis indicated that this phage was a member of subfamily *Autographivirinae*, family *Podoviridae*, and order *Caudovirales*. However, PHB01 did not belong to any genera previously identified in subfamily *Autographivirinae*, and should be assigned into a new genus. In addition, PHB01 displayed good ability on killing most of *P. multocida* type D strains, and it also had good therapeutic effect on the infections caused by *P. multocida* type D. This phage has a potential application against *P. multocida* capsular type D.

## Figures and Tables

**Figure 1 viruses-11-00086-f001:**
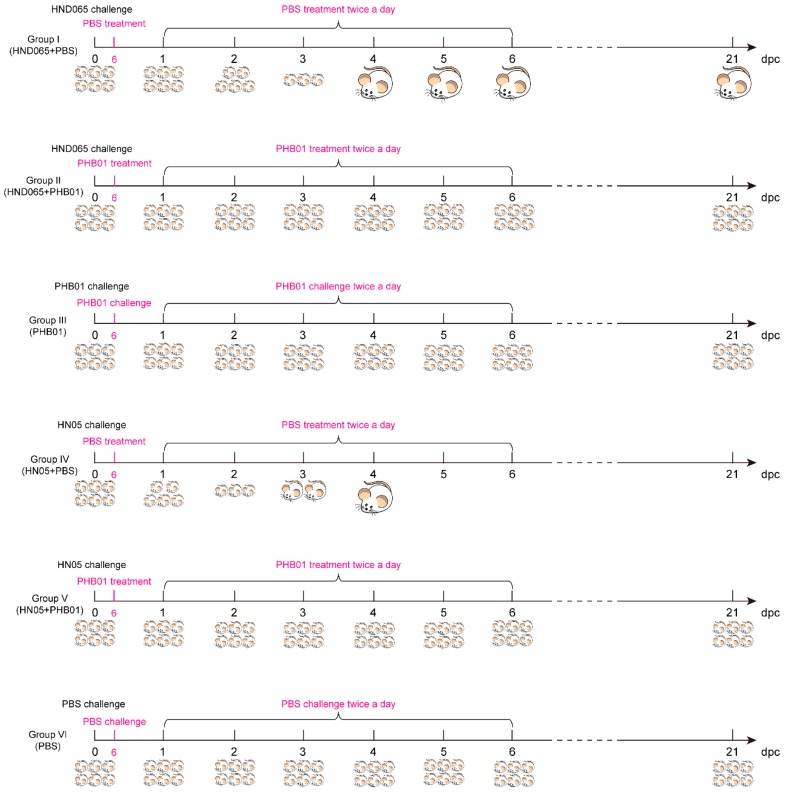
Experimental scheme for the evaluation of PHB01 treatment efficacy in mice infected with *P. multocida* type D. Each of the mice in the *P. multocida*-challenged groups were given a treatment of PHB01 at 10^8^ PFU (Groups II and V) or PBS (Groups I and IV) at 6 h following bacterial inoculation, and then twice a day for five days.

**Figure 2 viruses-11-00086-f002:**
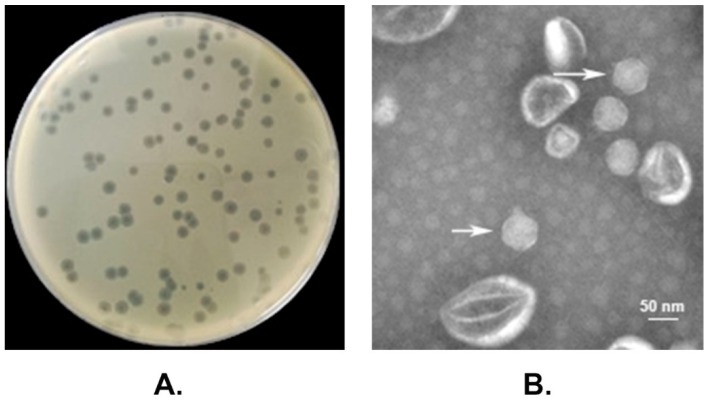
Morphological characteristics of phage PHB01. (**A**) Plaques of phage PHB01 on *Pasteurella multocida* HND065; (**B**) Transmission electron micrograph of phage PHB01 (marked with white arrows).

**Figure 3 viruses-11-00086-f003:**
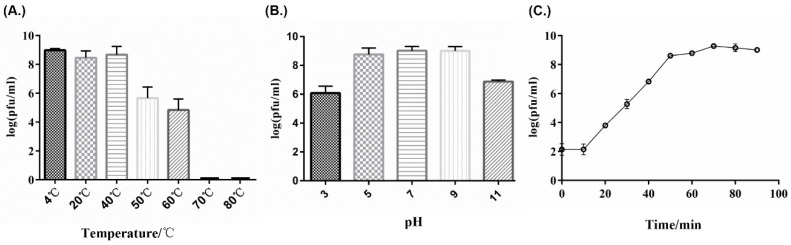
Biological properties of phage PHB01. (**A**) Sensitivity to temperature variations; (**B**) Sensitivity to pH variations; (**C**) One-step growth curve.

**Figure 4 viruses-11-00086-f004:**
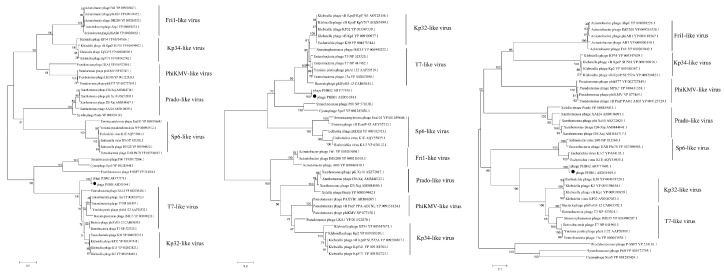
Phylogenetic tree analysis based on the alignments of amino acid sequences of the major capsid protein ((**A**); Protein ID: ASD51044.1), the DNA polymerases ((**B**); Protein ID: ASD51034.1), and the RNA polymerases ((**C**); Protein ID: ASD51019.1) of the *Autographivirinae* subfamily phages from GenBank. Phage PHB01 is indicated by the black circle. The evolutionary trees were constructed using the neighbor-joining method with the Poisson correction model. One thousand bootstrap repetitions were performed. Sequences were aligned by the ClustalW package carried by MEGAX [35].

**Figure 5 viruses-11-00086-f005:**
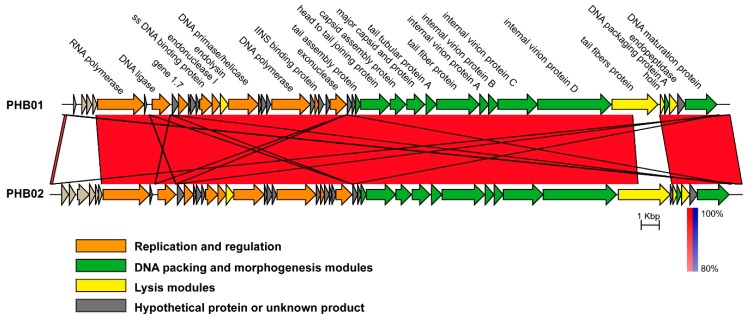
A co-linearity comparison diagram of the genomic organization at the nucleotide level between *Pasteurella* phages PHB01 and PHB02. The figure was generated via Easyfig v.2.0. The color code refers to the BLASTn identity of those regions between genomes. Arrows represent putative CDSs encoded by different genomes.

**Figure 6 viruses-11-00086-f006:**
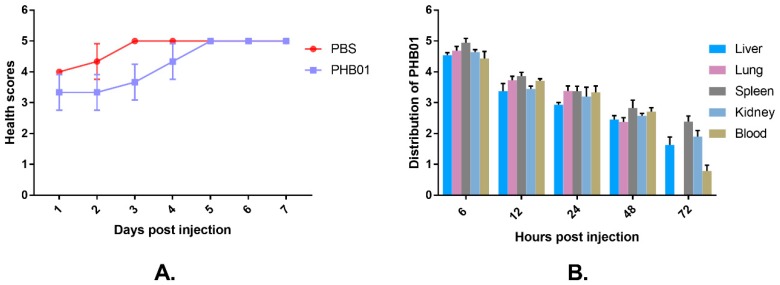
Safety test of PHB01 on mice. (**A**) Health scores given to the mice in different groups; (**B**) Distribution of phage PHB01 on the main organs of the mice. Data are presented as the mean ± SD.

**Figure 7 viruses-11-00086-f007:**
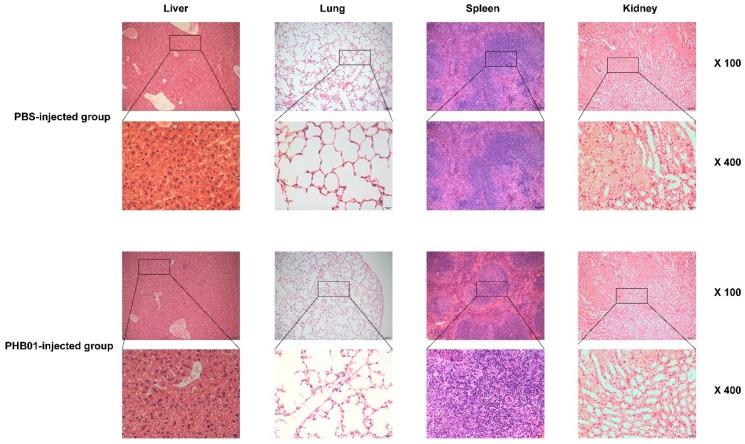
Histopathological analysis of the main organs collected from the mice received a challenge of phage PHB01 and/or PBS. The liver, spleen, kidney, and lungs were fixed with 4% formalin. Tissue sections were stained with hematoxylin and eosin or toluidine blue.

**Figure 8 viruses-11-00086-f008:**
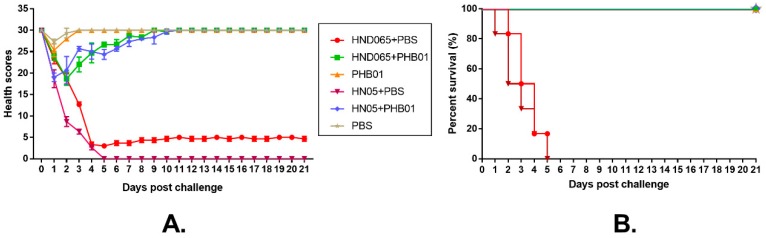
Protective effects of the phage PHB01 in mice challenged with wild-type *P. multocida* strain HND065 and HN05. (**A**) Health scores given to the mice in different groups. The total score for the health status of each group was recorded at least three times per day. The data are expressed as the mean ± SD; (**B**) Survival curve of the mice in each of the groups during the experiment.

**Table 1 viruses-11-00086-t001:** Host range of phage PHB01 ^1^.

Strain	Strain (Genotype)	Isolated Locations	Efficiency of Plating (EOP)
1	*Pasteurella multocida* D strain HND065	Henan, China	1
2	*Pasteurella multocida* D	Guangdong, China	<0.01
3	*Pasteurella multocida* D	Hubei, China	1.1
4	*Pasteurella multocida* D	Anhui, China	0.66
5	*Pasteurella multocida* D	Henan, China	1.33
6	*Pasteurella multocida* D	Hubei, China	1
7	*Pasteurella multocida* D	Hubei, China	0.5
8	*Pasteurella multocida* D	Hubei, China	0.06
9	*Pasteurella multocida* D	Guangdong, China	<0.01
10	*Pasteurella multocida* D	Hubei, China	<0.01
11	*Pasteurella multocida* D	Hubei, China	0.84
12	*Pasteurella multocida* D	Anhui, China	0.83
13	*Pasteurella multocida* D	Fujian, China	<0.001
14	*Pasteurella multocida* D	Guangdong, China	0.25
15	*Pasteurella multocida* D	Hubei, China	1.5
16	*Pasteurella multocida* D	Shanxi, China	0.58
17	*Pasteurella multocida* D	Shanghai, China	0.5
18	*Pasteurella multocida* D	Shanxi, China	0.75
19	*Pasteurella multocida* D	Hubei, China	0.5
20	*Pasteurella multocida* D	Hubei, China	0.66
21	*Pasteurella multocida* D	Hubei, China	<0.001
22	*Pasteurella multocida* D	Guangdong, China	<0.001
23	*Pasteurella multocida* D	Guangdong, China	-
24	*Pasteurella multocida* D	Anhui, China	-
25	*Pasteurella multocida* D	Shanxi, China	-
26	*Pasteurella multocida* D	Guangdong, China	-
**27**	***Pasteurella multocida* D**	**Unknown**	-
**28**	***Pasteurella multocida* D**	**Unknown**	-
**29**	***Pasteurella multocida* D**	**Unknown**	-
**30**	***Pasteurella multocida* D**	**Unknown**	-
**31**	***Pasteurella multocida* D**	**Unknown**	-
**32**	***Pasteurella multocida* D**	**Unknown**	-
**33**	***Pasteurella multocida* D**	**Unknown**	-
**34**	***Pasteurella multocida* D strain HN06**	**Hainan, China**	-
**35**	***Pasteurella multocida* D**	**Hubei, China**	-
**36**	***Pasteurella multocida* D**	**Guangdong, China**	-
**37**	***Pasteurella multocida* D**	**Shanxi, China**	-
38	*Pasteurella multocida* A	Beijing, China	-
39	*Pasteurella multocida* A	Hubei, China	-
40	*Pasteurella multocida* A	Hubei, China	-
41	*Pasteurella multocida* A	Hunan	-
42	*Pasteurella multocida* A	Sichuan	-
43	*Pasteurella multocida* A	Sichuan	-
44	*Pasteurella multocida* A	Guangdong	-
45	*Pasteurella multocida* A	Guangdong	-
46	*Pasteurella multocida* A strain HB01	Hubei	-
47	*Pasteurella multocida* A strain HB03	Hubei	-
48	*Pasteurella multocida* F strain HN07	Henan	-
49	*Escherichia coli* DH5α		-
50	*Salmonella enterica* serovar *typhimurium* ATCC14028		-
51	*Salmonella enterica* serovar *choleraesuis*		-
52	*Bordetella bronchiseptica*		-

^1^ The EOP values were determined by calculating the ratio of plaque-forming units (PFUs) of each phage-susceptible strain to the PFUs of indicator strain (*P. multocida* HND065); toxigenic capsular type D strains were marked in bold; (-) indicates that no plaques were observed.

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
