# Peer review of "Isolation of a T7-Like Lytic Pasteurella Bacteriophage vB_PmuP_PHB01 and Its Potential Use in Therapy against Pasteurella multocida Infections"

_viruses, 2019, doi:10.3390/v11010086_

Reviewer 1 Report

The work describes the characterisation of the phage PHB01 and its potential use as a therapeutic agent. Clearly there is a large amount of work that has gone into this paper. The physical characterisation and the  use of the phage as therapeutic are presented in a concise manner and provide the potential to move the field forward. However, as it stand there are currently a number of issues with manuscript as it stands, in particular bioinformatics analysis. These need to be addressed prior to its publication.

Throughout the manuscript, there are details missing about how techniques were carried e.g. the method used for DNA extraction. In particular, details are needed for all the bioinformatics tools used, version and parameters. How genes were annotated with a function, what cut-off values were used.

The title itself is misleading; the phage is 96% similar (authors words) to an existing phage. Based on current standards this is a strain of already isolated species …. So not novel in that sense.

The alignment of distantly related genomes and comparison in phylogeny is not informative (Figure 5A). The analysis in figure 5B is also dubious, see comments below for details. Based on the annotation the new phage has many genes in common with T7 and its relatives. A homology-based analysis should be carried out to determine how many, many packages are available to do this ( orthomcl, Prokka, get_homologues). This can then be used to construct informative analysis to determine its phylogenetic placement – currently only 1 gene has been used (from an entire genome) against a select number of phages. Without justification of why these phages were selected.

Minor comments

L25 P multocida shoudnt be abbreviate in the abstract on first use

L43 – wrong tense, happened

L47 -51. This is a worrying description of lytic and “lysogenic” phage. The phage is “temperate” not lysogenic. The host is lysogenic, not the phage. Furthermore, temperate phage can cause lysis of their hosts, the way this description is currently written does not make this obvious. Please rewrite to demonstrate the fundamental understanding of phage biology

L72 of sewage

 L72 here and elsewhere. Space between units and numbers. 5  ml

L75 insert “pore size.”

L77 here and elsewhere is the % w/v  or v/v etc. Please specify

L83 insert pore size. And on any other occasion that is required about the size of pores in a membrane

L94 this sentence is clumsily written 

L120, please provide  a method for extraction

L122 provide details of parameters used for assembly. The version number of software used. What was the depth of sequencing, how were reads trimmed? Please submit raw fastq files to a public database

L127 what is meant by WGS here?

L131 –how was sequence comparison done

L132 – details of how phylogenetic analysis was carried out are needed

Line 140 how were the phage purified?

Figure 2B the image of the phage is of poor quality . The scale should be marked in the image as well as the legend . the number of measurement is also very small

Line 188 – please give details of the log fold drop in the text

Figure 3C , why was eclipse period not calculated ?

Line 209- how was the 223 bp terminal redundancy found ? How were functions assigned to a gene ? What programs/tools. What cutoff values were used

L218 – how was identity determined, what tool ? If an ANI calculator was not used, please use one.

L219 – homology cannot be quantified. Remove significant or replace homology with identity

L222 how different?

Figure 4 How was the figure made? What were blastn cutoffs used? What is the blue shading in the figure legend for. What were the pairwise comparisons made?

Line 232 – how was the analysis produced? If figure 5B1  was produced by comparing the entire genome in alignment against other phages, then this an entirely in appropriate analysis to be carried out . Details are required

Figure 5B – The gene that was used to construct this tree should be stated (Accession numbers ) and the accession numbers of other genes in the tree from other species.  Based on the details provided this analysis appears to be incorrect

There is no sequence in PHB01 that has the description “major capsid” ! this should be amended. It is not at all clear why the sequences presented in the tree were chosen ? There are well over a 100 phages that have T7like major capsid proteins, these should be included in the tree. What was the outgroup used for this tree ? The cyanophages P60 or Syn5 would be useful starting points . The tree appears to be midpoint branched .   Having used the major capsid of T7 and tried to repeat the analysis, very different results are found.  Details of this analysis need to be provided with a greater number of sequences used in the analysis.

Further analysis with a 2nd gene such as a terminase or DNA polymerase is also required, as this analysis does not support the conclusions made

Line 218 death conditions in strange term . Did the mice die?

Line 324- It is not clear how Figure 5A was produced, so these conclusions are dubious

Line 337 – 338. Without the doubt this new phage and itsPHB01 are closely related and part of the Podoviridae. What is not clear is how similar they are to other phage. They may well represent a new genus but no data is presented to support this claim as it stands  

Table S1 “T7 RNA polymerase [synthetic construct] “ is not a

Author Response

L25 P multocida shoudnt be abbreviate in the abstract on first use

Response: Thank you for your suggestion. The place has been revised as suggested. Please see line 25.

L43 – wrong tense, happened

Response: Thank you for your suggestion. The tense has been revised as suggested. Please see line 43.

L47 -51. This is a worrying description of lytic and “lysogenic” phage. The phage is “temperate” not lysogenic. The host is lysogenic, not the phage. Furthermore, temperate phage can cause lysis of their hosts, the way this description is currently written does not make this obvious. Please rewrite to demonstrate the fundamental understanding of phage biology

Response: Thank you for your comment. The sentence has been removed. Please see lines 48-49.

L72 of sewage

Response: Thank you for your comment. The place has been revised as suggested. Please see line 68.

 L72 here and elsewhere. Space between units and numbers. 5  ml

Response: Thank you for your suggestion. We are sorry for our carelessness. We have checked the whole manuscript and revised all the places with the similar problem. Please see lines 69, 71, 72, etc. 

L75 insert “pore size.”

Response: Thank you for your suggestion. We have revised the contents as suggested. Please see lines 68, 71, 80, etc.

L77 here and elsewhere is the % w/v  or v/v etc. Please specify

Response: Thank you for your suggestion.  We have revised the contents as suggested. Please see lines 73 and 75.

L83 insert pore size. And on any other occasion that is required about the size of pores in a membrane

Response: Thank you for your suggestion. We have revised the contents as suggested. Please see lines 68, 71, 80, etc.

L94 this sentence is clumsily written 

Response: Thank you for your suggestion. We have re-written the sentence. Please see lines 90-92. 

L120, please provide  a method for extraction

Response: Thank you for your suggestion. We have provided  a method for extraction. Please see lines 120-127. 

L122 provide details of parameters used for assembly. The version number of software used. What was the depth of sequencing, how were reads trimmed? Please submit raw fastq files to a public database

Response: Thank you for your suggestion. We have provided the required information. Please see lines 128-136, 139-140. 

L127 what is meant by WGS here?

Response: Thank you for your comment. We have re-written the sentence. Please see lines 141-142.

L131 –how was sequence comparison done

Response: The sequence comparison was done by using software EasyFig v. 2.0. Please see lines 140-141.

L132 – details of how phylogenetic analysis was carried out are needed

Response: Thank you for your suggestion. Details for the phylogenetic analysis have been provided. Please see lines 141-142, 231-233. 

Line 140 how were the phage purified?

Response: Thank you for your suggestion. The phages were purified by CsCl gradient ultra centrifugation. Please see lines 82-83.  

Figure 2B the image of the phage is of poor quality . The scale should be marked in the image as well as the legend . the number of measurement is also very small

Response: Thank you for your suggestion. We have replaced figure 2B with a high quality one. Please see new figure 2B.

Line 188 – please give details of the log fold drop in the text

Response: Thank you for your suggestion. The log fold drop has been added. Please see line 197. 

Figure 3C , why was eclipse period not calculated ?

Response: Thank you for your comments. We have calculated the eclipse period in our manuscript. Please see line 201. 

Line 209- how was the 223 bp terminal redundancy found ? How were functions assigned to a gene ? What programs/tools. What cutoff values were used

Response: Thank you for your comments. The terminal sequences of the virus genome were determined by a modified statistical method (Li, S., Fan, H., An, X., Fan, H., Jiang, H., Chen, Y., Tong, Y. 2014. Scrutinizing virus genome termini by high-through put sequencing. PLoS One. 9:e85806.). We have added the contents in the method section. See lines 135-136. 

As we have described in the submission (lines 136-138), the genes were predicted and annotated using the common online tools widely used for genome sequence annotation. Their version number and parameters have been also given in the manuscript (lines 136-138).

L218 – how was identity determined, what tool ? If an ANI calculator was not used, please use one.

Response: The DNA identity was given automatically by the sequence comparison tool EasyFig v. 2.0.

L219 – homology cannot be quantified. Remove significant or replace homology with identity

Response: According to the suggestions of the other reviewers, we have removed the contents and have rewritten the paragraph. Please see lines 234-242 . 

L222 how different?

Response: Thank you for your comment. We have mentioned the difference in our manuscript. See lines 240-242.

Figure 4 How was the figure made? What were blastn cutoffs used? What is the blue shading in the figure legend for. What were the pairwise comparisons made?

Response: The figure was generated using the sequence comparison tool EasyFig v. 2.0. The pairwise comparison was made by blast carried on the software (EasyFig v. 2.0). We have mentioned the contents in our method section. See lines 140-141.

Line 232 – how was the analysis produced? If figure 5B1  was produced by comparing the entire genome in alignment against other phages, then this an entirely in appropriate analysis to be carried out . Details are required

Figure 5B – The gene that was used to construct this tree should be stated (Accession numbers ) and the accession numbers of other genes in the tree from other species.  Based on the details provided this analysis appears to be incorrect

There is no sequence in PHB01 that has the description “major capsid” ! this should be amended. It is not at all clear why the sequences presented in the tree were chosen ? There are well over a 100 phages that have T7like major capsid proteins, these should be included in the tree. What was the outgroup used for this tree ? The cyanophages P60 or Syn5 would be useful starting points . The tree appears to be midpoint branched .   Having used the major capsid of T7 and tried to repeat the analysis, very different results are found.  Details of this analysis need to be provided with a greater number of sequences used in the analysis.

Further analysis with a 2nd gene such as a terminase or DNA polymerase is also required, as this analysis does not support the conclusions made

Response: Thank you for your comment. First of all, the phylogenetic analysis was performed by MEGAX, a software which is commonly used for phylogenetic analysis (Kumar, S.; Stecher, G.; Li, M.; Knyaz, C.; Tamura, K. MEGA X: Molecular Evolutionary Genetics Analysis across Computing Platforms. Mol Biol Evol 2018, 35, 1547-1549), and the figures were exported automatically by the software. We have mentioned such method in lines 131-132, and 238-257 in our initial submission, and also lines 141-142, and 231-233 in our reversion (the version). Then, we have removed the original figure 5A, and re-generated the phylogenetic trees using amino acid sequences of the major capsid protein (the new Figure 5A), the DNA polymerases (Figure 5B), and the RNA polymerases (Figure 5C) according to your suggestions.  

Line 218 death conditions in strange term . Did the mice die?

Response: Thank you for your comment. We have re-written the sentence. Please see lines 267-268.

Line 324- It is not clear how Figure 5A was produced, so these conclusions are dubious

Response: Thank you for your comment. The phylogenetic analysis was performed by MEGAX, a software which is commonly used for phylogenetic analysis (Kumar, S.; Stecher, G.; Li, M.; Knyaz, C.; Tamura, K. MEGA X: Molecular Evolutionary Genetics Analysis across Computing Platforms. Mol Biol Evol 2018, 35, 1547-1549), and the figure5A were exported automatically by the software. We have also re-written the contents in the discussion section. Please see lines 302-304. 

Line 337 – 338. Without the doubt this new phage and its PHB01 are closely related and part of the Podoviridae. What is not clear is how similar they are to other phage. They may well represent a new genus but no data is presented to support this claim as it stands  

Response: Thank you for your comment. We have re-written the contents in the discussion section and have removed the improper statement. Please see lines 302-314.

Table S1 “T7 RNA polymerase [synthetic construct] “ is not a

Response: Thank you for your comment. We have revised the place. Please see Table S1.

Reviewer 2 Report

In the Manuscript entitled “ Isolation of a novel T7-like lytic Pasteurella bacteriophage vB_PmuP_PHB01 and its potential use in therapy against Pasteurella multocida infections” Chen and Colleagues describe a newly discovered lytic bacteriophage specific to  P. multocida type D strains. The phage has been thoroughly characterized through classic approaches (host range, sensitivity to physical agents, One-Step growth curve, TEM) and its genome was obtained and analyzed. Finally, the ability of the phage to protect animals from death has been assessed in a mouse model. The Manuscript is well written and organized, methods appear appropriate and results are clearly presented. I have therefore just two main suggestion for Authors to improve the paper and a number of minor comments mostly of editorial style:

The description of the protocol  for the determination of EOP appears to be missing. It should be at least briefly described in the Materials and Methods section

The exact length of direct terminal repeats (223 bp) seems to be predicted by using just a bioinformatics approach. Even if I realize that the determination of this parameter through wet-lab techniques is rather cumbersome, this result should be at least double-checked by using other bioinformatics tools (e.g. PhageTerm, etc).

Minor comments:

Lines 80-81. I would use here the term modified SM buffer, just because gelatin is not present in the original recipe.
Line 82. The acceleration due to gravity(g) should be reported in italic characters, e.g. in the same way done at line 102.
Line 85.  “and the isolates were” could be transformed to “and the phage suspensions were”, to better describe the obtained sample.
Line 104. Considering that in Figure 3c time is reported in minutes, I would suggest to use the same unit of measurement here (i.e. t = 0 min instead of t = 0 s).
Line 111-122. As probably well-known by the Authors, the official taxonomic nomenclature system for the Salmonella genus is rather complicated and sometimes ambiguous. Given this fact I suggest the Authors to double-check if the naming of Salmonella typhimurium and Salmonella choleraesuis used here is according to the most recent taxonomic classification.
Line 144. Status, misspelled.
Lines 157-158. 107 should be 10 to the 7, 104 should be 10 to the 4 and 109 should be 10 to the 9. In addition, please specify the actual PFU (1.0 x 10 to the 8, as specified at line 259?) used for challenging and not just the concentration (i.e. 10 to the 9 PFU/ml)
Line 169. 109 should be 10 to the 9
Line 195. PHB02 should be PHB01? In addition “(A) temperature; (B) pH;” could be substituted with “(A) sensitivity to temperature variations; (B) sensitivity to pH variations;”
Line 202. Salmonella spp instead of just Salmonella.
Line 230. “encoded by different” instead of “encoded y different”
Line 247. “28,066” instead of “28066”
Line 259. 1.0x10to the 8.
Line 259-260. The meaning of this sentence is not very clear to this reviewer. Why the statement on injections is repeated twice? Please, rephrase it.
Line 270. There is an inconsistency between the label of y axes of Fig 6B and the corresponding legend. PHB02 should be PHB01 at this line?
Line 293. “phage” instead of “phages”

Author Response

Lines 80-81. I would use here the term modified SM buffer, just because gelatin is not present in the original recipe.

Response: Thank you for your suggestions. We have reversed the content. Please see lines 77-78. 
Line 82. The acceleration due to gravity(g) should be reported in italic characters, e.g. in the same way done at line 102.

Response: Thank you for your suggestions. We have reversed the contents. Please see line 79. 

Line 85.  “and the isolates were” could be transformed to “and the phage suspensions were”, to better describe the obtained sample. 

Response: Thank you for your suggestions. We have reversed the content. Please see line 82. 

Line 104. Considering that in Figure 3c time is reported in minutes, I would suggest to use the same unit of measurement here (i.e. t = 0 min instead of t = 0 s).

Response: Thank you for your suggestions. We have reversed the content. Please see line 101. 

Line 111-122. As probably well-known by the Authors, the official taxonomic nomenclature system for the Salmonella genus is rather complicated and sometimes ambiguous. Given this fact I suggest the Authors to double-check if the naming of Salmonella typhimurium and Salmonella choleraesuis used here is according to the most recent taxonomic classification.

Response: Thank you for your suggestions. We have reversed the content. Please see lines 108, 109, and Table 1. 

Line 144. Status, misspelled.

Response: We are sorry for our carelessness. We have changed the spell. Please see line 154.

Lines 157-158. 107 should be 10 to the 7, 104 should be 10 to the 4 and 109 should be 10 to the 9. In addition, please specify the actual PFU (1.0 x 10 to the 8, as specified at line 259?) used for challenging and not just the concentration (i.e. 10 to the 9 PFU/ml)

Response: Thank you for your suggestions. We have reversed the contents. Please see lines 152, 167, 168, 169, 170, 249, etc.

Line 169. 109 should be 10 to the 9

Response: Thank you for your suggestions. We have reversed the contents. Please see lines 152, 167, 168, 169, 170, 249, etc.

Line 195. PHB02 should be PHB01? In addition “(A) temperature; (B) pH;” could be substituted with “(A) sensitivity to temperature variations; (B) sensitivity to pH variations;”

Response: Thank you for your suggestions. We have reversed the contents. Please see lines 204-205.

Line 202. Salmonella spp instead of just Salmonella.

Response: Thank you for your suggestions. We have reversed the content. Please see line 212.

Line 230. “encoded by different” instead of “encoded y different”

Response: Thank you for your suggestions. We have reversed the content. Please see line 247.

Line 247. “28,066” instead of “28066”

Response: Thank you for your suggestions. We have removed this section according to the other re-viewers' comment and re-written the section. Please see lines 231-233.

Line 259. 1.0x10to the 8.

Response: Thank you for your suggestions. We have reversed the content. Please see line 249.

Line 259-260. The meaning of this sentence is not very clear to this reviewer. Why the statement on injections is repeated twice? Please, rephrase it.

Response: Thank you for your suggestions. We have rewritten the sentence. Please see line 249.

Line 270. There is an inconsistency between the label of y axes of Fig 6B and the corresponding legend. PHB02 should be PHB01 at this line?

Response: We are sorry for our carelessness. We have changed the name. Please see line 258. Thank you.

Line 293. “phage” instead of “phages”

Response: Thank you for your suggestions. We have reversed the content. Please see line 280.

Reviewer 3 Report

The paper contains the new experimental material concerning the characterization of a new Pasteurella bacteriophage, the testing of its curative effect and biological safety. The experiments are well described and presented.

Major critical notes deal with the bioinformatic analysis and illustrations.

1) The diagram of the genome directed from right to left is puzzling (Fig.4)

2) The graphic comparison of PHB01 genome with the genomes of Pasteurella phages obviously belonging to different taxonomic  family has very few sense (Fig.4), the same for phylogenetic tree Fig 5A - it reflects the assignment of Pasteurella phages to distant taxa.

3) The proposition of the new viral genus should be based on not only nucleotide similarity comparison, but on notable genomic features distinguishing the discussed phages from other known Autographivirinae. The examples are: positioning of the polymerase gene in the genome, structures of adsorption and lysis modules, promotor arrangement etc. None of such analysis is offered in the paper.

4) The phylogenetic relations between phage PHB01 and other T7virus phages are not clear. Fig.5B shows the certain distance from KP34, Fri1, SP6 and KP32. But what about other T7virus representatives? There exist much more of them rather than Kp1 and T7 shown in the tree. A more complete phylogenetic analysis should be performed in order to better assign the genome of PHB01. At least the analysis should include those Yersinia, Klebsiella and Escherichia phages revealing the highest similarity in certain genes (Table S1)

Other:

The statement in lines 330-333 is incorrect. The difference in the structure of polysacchrides of different capsular types is not a proper explanation to the fact that T7like phages infecting type B strains are not known yet. 

Line 230 "CDSs encoded y different genomes" - in? by?

Overall the manuscript is good, but requires some addtional effort in the proof of a new genus proposition.

Author Response

Major critical notes deal with the bioinformatic analysis and illustrations.

1) The diagram of the genome directed from right to left is puzzling (Fig.4)

Response: Thank you for your suggestions. We have re-generated the figure according to the comments from both you and the other re-viewers. Please see the new figure (figure 5) in the re-submission.

2) The graphic comparison of PHB01 genome with the genomes of Pasteurella phages obviously belonging to different taxonomic  family has very few sense (Fig.4), the same for phylogenetic tree Fig 5A - it reflects the assignment of Pasteurella phages to distant taxa.

Response: Thank you for your suggestions. We have removed the original figure 5A, and figure 4 in the initial submission. We have also re-written the related section (lines 226-242), and regenerated the phylogenetic trees (Figure 4), and the whole sequence comparison (figure 5) in the revised version.

3) The proposition of the new viral genus should be based on not only nucleotide similarity comparison, but on notable genomic features distinguishing the discussed phages from other known Autographivirinae. The examples are: positioning of the polymerase gene in the genome, structures of adsorption and lysis modules, promotor arrangement etc. None of such analysis is offered in the paper.

Response: Thank you for your suggestions. We have removed the improper states in the discussion section (lines 302-314) in the revised version.

4) The phylogenetic relations between phage PHB01 and other T7virus phages are not clear. Fig.5B shows the certain distance from KP34, Fri1, SP6 and KP32. But what about other T7virus representatives? There exist much more of them rather than Kp1 and T7 shown in the tree. A more complete phylogenetic analysis should be performed in order to better assign the genome of PHB01. At least the analysis should include those Yersinia, Klebsiella and Escherichia phages revealing the highest similarity in certain genes (Table S1)

Response: Thank you for your suggestions. We have re-done the phylogenetic analysis accordingly (lines 231-233) and regenerated the phylogenetic trees (Figure 4) in the revised version. Also, we have removed all states involved in "novel" from the manuscript.

Other:

The statement in lines 330-333 is incorrect. The difference in the structure of polysacchrides of different capsular types is not a proper explanation to the fact that T7like phages infecting type B strains are not known yet. 

Response: Thank you for your suggestions. We have removed the improper address and have rewritten the contents in the discussion section (lines 302-314) in the revised version.

Line 230 "CDSs encoded y different genomes" - in? by?

Response: We are sorry for our carelessness and we have changed the incorrect write. Please see line 247.

Round  2

Reviewer 1 Report

The updated manuscript is much improved with most points addressed. It adds to the increasing research on phage therapy and demonstrates the safe application of the phage in a mouse mode.  However, there are still a number of issues . The figures are of poor quality and hard to read/fuzzy. This might be a conversion issue given most Figures seemed to have this issue.  The analysis and annotation of the genome still needs to be improved in  terms of the detail provided- points are listed below

Line 128-129 – the library prep is still not stated and needs to be

Line 130-133- the details of QC are now provided. But please state the software used for QC

Line 141 version is not included and should be

Line 193- the red arrows and I assume scale bar are not visible in the Figure 2 B

Line 201 – It is not clear from Figure 3C how the eclipse period was calculated. The much improved materials and methods also do not specify how this was done. The details of how this was determined should be included.

Figure 4 the method of tree construction is not included and needs to be eg the model of evolution that was used. NJ, Parsimony, ML . Details of how sequences were aligned should also be included in the methods- what tool

Line 234 –The sentence does not make sense .I think one of the PHB01 should be PHB02

Line 236 –This Figure of 96% is only over 91% of the genome. Please use an ANI calculator  to report an accurate  average nucleotide identity between the genomes. (http://enve-omics.ce.gatech.edu/ani/), rather than a blastn identity

Figure 4. These trees are unreadable. I cannot repeat the phylogenetic analysis that has been carried out . There are a number of issues that are present:

1)      The trees are not completely readable – this is possibly an issue with conversion . But needs to be corrected

2)      It is unclear what genes have been used to construct the phylogeny – accession numbers would help. There is still not any annotation of major capsid protein the genbank file –this maybe an issue with updating the file. However, looking at the table it is not clear which gene was used. ORF 30 is listed as major capsid protein – in the genbank file there is no ORF 30 (which doesn’t help) or major capsid protein. Going off the co-ordinates then the gene annotated as gp9 (genbank file) has been used- I think ? The format of the table makes it difficult  to tell

A BLAST analysis reveals that gene annotated as gp10A(genbank) is the likely major capsid protein. Again I cannot tell if this a formatting issue with the supplementary table S1. But having different annotations in the Table to the genbank file is not helpful and confusing to the reader. I am unclear what gene has been used for analysis and what it has been compared too .Please provide accession numbers of proteins for genes that have been compared in the legends of the Figures

3)      It is not clear if an outgroup was used, they still appear to midpoint branched. If this is the case then please state in the legend.

After receipt of the updated Figures, they are now readable , with accession numbers included in the tree. This adds important detail for interpretation and for others to expand/repeat this analysis if they wished to do so. The points around the annotation and supplementary file still stand – in the text the first tree refers to the  major capsid protein. But not annotated as such in the Genbank file. It is called gp10A and “ capsid and scaffold protein”– which unless people are familiar with T7 is a meaningless annotation.

The details on how the tree was constructed, what parameters were used ,what model of evolution (was this tested), how was the alignment constructed (what alignment tool , stating MEGA  is not sufficient, this contains a collection of tools ) . Having tried to repeat the analysis for the DNA polymerase, I cannot construct a similar phylogeny based on the limited details provided.  I have no doubt the phage is a member of the Autographivirinae as the authors state, but its exact phylogeny within this group is not clear based on the details that are provided.

The Figure contains groups of phages, with the names such as T7-like virus. They should not be referred to in this way, the current nomenclature for a genus is T7virus, Fri1virus etc , the “like” should be removed.

Figure 5 is of poor quality, the text is unreadable. Again it might be an issue with the conversion ?

The updated Figure 5 that was sent, is a lot clearer to read. The only issue with this updated Figure is the blue bar in the Legend, that does not seem to be needed

Line 327- The morphological data supports its identification as Podoviridae. The genetic data supports it fits within the Autographivirinae. The figures cannot be read, it is possible that is a new genus. However, the figures and description need to be improved to confirm this.

Author Response

Line 128-129 – the library prep is still not stated and needs to be

Response: The required information has been added. See lines 132-133. Thank you.

Line 130-133- the details of QC are now provided. But please state the software used for QC

Response: We have added the software as well as its version number used for QC control. See line 134. Thank you.

Line 141 version is not included and should be

Response: Actually we have mentioned the version of the software (v.2.0). See line 145. Thank you.

Line 193- the red arrows and I assume scale bar are not visible in the Figure 2 B

Response: We have re-generated the figure. See the updated figure 2B. Thank you.

Line 201 – It is not clear from Figure 3C how the eclipse period was calculated. The much improved materials and methods also do not specify how this was done. The details of how this was determined should be included.

Response: The required information has been added in the Method section. Please see lines 104-106. Thank you.

Figure 4 the method of tree construction is not included and needs to be eg the model of evolution that was used. NJ, Parsimony, ML . Details of how sequences were aligned should also be included in the methods- what tool

Response: The required information has been added. Please see lines 237-239. Thank you.

Line 234 –The sentence does not make sense .I think one of the PHB01 should be PHB02

Response: We have re-written the sentence. Please see lines 240-241. Thank you.

Line 236 –This Figure of 96% is only over 91% of the genome. Please use an ANI calculator  to report an accurate  average nucleotide identity between the genomes. (http://enve-omics.ce.gatech.edu/ani/), rather than a blastn identity

Response: The required information has been added as suggested. Please see lines 240-241. Thank you.

Figure 4. These trees are unreadable. I cannot repeat the phylogenetic analysis that has been carried out . There are a number of issues that are present:

1)      The trees are not completely readable – this is possibly an issue with conversion . But needs to be corrected

Response: A more clear vectorgraph has been re-generated and put in the manuscript. You can enlarge it to read it clearly. Please see the new figure 4. Thank you. 

2)      It is unclear what genes have been used to construct the phylogeny – accession numbers would help. There is still not any annotation of major capsid protein the genbank file –this maybe an issue with updating the file. However, looking at the table it is not clear which gene was used. ORF 30 is listed as major capsid protein – in the genbank file there is no ORF 30 (which doesn’t help) or major capsid protein. Going off the co-ordinates then the gene annotated as gp9 (genbank file) has been used- I think ? The format of the table makes it difficult  to tell 

 A BLAST analysis reveals that gene annotated as gp10A(genbank) is the likely major capsid protein. Again I cannot tell if this a formatting issue with the supplementary table S1. But having different annotations in the Table to the genbank file is not helpful and confusing to the reader. I am unclear what gene has been used for analysis and what it has been compared too .Please provide accession numbers of proteins for genes that have been compared in the legends of the Figures

Response: ID numbers for the proteins used for the generation of the trees have been given. Please see lines 235 and 236. Thank you. 

3)      It is not clear if an outgroup was used, they still appear to midpoint branched. If this is the case then please state in the legend.

After receipt of the updated Figures, they are now readable , with accession numbers included in the tree. This adds important detail for interpretation and for others to expand/repeat this analysis if they wished to do so. The points around the annotation and supplementary file still stand – in the text the first tree refers to the  major capsid protein. But not annotated as such in the Genbank file. It is called gp10A and “ capsid and scaffold protein”– which unless people are familiar with T7 is a meaningless annotation.

Response: ID numbers for the proteins used for the generation of the trees have been given. Please see lines 235 and 236. Thank you. 

The details on how the tree was constructed, what parameters were used ,what model of evolution (was this tested), how was the alignment constructed (what alignment tool , stating MEGA  is not sufficient, this contains a collection of tools ) . Having tried to repeat the analysis for the DNA polymerase, I cannot construct a similar phylogeny based on the limited details provided.  I have no doubt the phage is a member of the Autographivirinae as the authors state, but its exact phylogeny within this group is not clear based on the details that are provided.

Response: Strategies for the generation of the trees have been given. Please see lines 237-239. Thank you. 

The Figure contains groups of phages, with the names such as T7-like virus. They should not be referred to in this way, the current nomenclature for a genus is T7virus, Fri1virus etc , the “like” should be removed.

Response: The word has been removed as suggested. Please see the new figure 4. Thank you. 

Figure 5 is of poor quality, the text is unreadable. Again it might be an issue with the conversion ?

The updated Figure 5 that was sent, is a lot clearer to read. The only issue with this updated Figure is the blue bar in the Legend, that does not seem to be needed

Response: The figure has been re-generated with high resolutions. Please see the new figure 5. Thank you. 

Line 327- The morphological data supports its identification as Podoviridae. The genetic data supports it fits within the Autographivirinae. The figures cannot be read, it is possible that is a new genus. However, the figures and description need to be improved to confirm this.

Response: Autographivirinae is a subfamily of the Podoviridae family. See our statement at line 195. Thank you.